# METAPIX: FEW-SHOT VIDEO RETARGETING

**Jessica Lee    Deva Ramanan    Rohit Girdhar**
Carnegie Mellon University
https://imjal.github.io/MetaPix

## ABSTRACT

We address the task of retargeting of human actions from one video to another. We consider the challenging setting where only a few frames of the target is available. The core of our approach is a conditional generative model that can transcode input skeletal poses (automatically extracted with an off-the-shelf pose estimator) to output target frames. However, it is challenging to build a universal transcoder because humans can appear wildly different due to clothing and background scene geometry. Instead, we learn to adapt – or *personalize* – a universal generator to the particular human and background in the target. To do so, we make use of *meta*-learning to discover effective strategies for on-the-fly personalization. One significant benefit of meta-learning is that the personalized transcoder naturally enforces temporal coherence across its generated frames; all frames contain consistent clothing and background geometry of the target. We experiment on in-the-wild internet videos and images and show our approach improves over widely-used baselines for the task.

## 1 INTRODUCTION

One of the hallmarks of human intelligence is the ability to imagine. For example, given an image of a never-before-seen person, one can easily imagine them performing different actions. To do so, we make use of years of experience watching humans act and interact with the world. We implicitly encode the rules of physical transformations of humans, objects, clothing and so on. Crucially, we effortlessly adapt or *retarget* those universal rules to a specific human and environment - a child on a playground will likely move differently than an adult walking into work. Our goal in this work is to develop models that similarly learn to generate human motions by specializing universal knowledge to a particular target human and target environment, given only a few samples of the target.

It is attractive to tackle such video generation tasks using the framework of generative (adversarial) neural networks (GANs). Past work has cast the core computational problem as one of conditional image generation where input source poses (automatically extracted with an off-the-shelf pose estimator) are transcoded into image frames (Balakrishnan et al., 2018; Siarohin et al., 2018; Ma et al., 2017). However, it is notoriously challenging to build generative models that are capable of synthesizing diverse, in-the-wild imagery. Notable exceptions make use of massively-large networks trained on large-scale compute infrastructure (Brock et al., 2019). However, modestly-sized generative networks perform quite well at synthesis of targeted domains (such as faces (Bansal et al., 2018) or facades (Isola et al., 2017)). A particularly successful approach to generating from pose-to-image is training of specialized – or *personalized* – models to particular scenes. These often require large-scale target datasets, such as 20 minutes of footage in a target lab setting (Chan et al., 2018)

The above approaches make use of personalization as an implicit but crucial ingredient, by *on-the-fly* training of a generative model tuned to the particular target domain of interest. Often, personalization is operationalized by fine-tuning a generic model on the specific target frames of interest. Our key insight is recasting personalization as an *explicit* component of a video-retargeting engine, allowing us to make use of meta-learning to *learn* how best to fine-tune (or personalize) a generic model to a particular target domain. We demonstrate that (meta)learning-to-fine-tune is particularly effective in the few-shot regime, where few target frames are available. From a technical perspective, one of our contributions is extending meta-learning to GANs, which is nontrivial because both a generator and discriminator need to be adversarially fine-tuned.

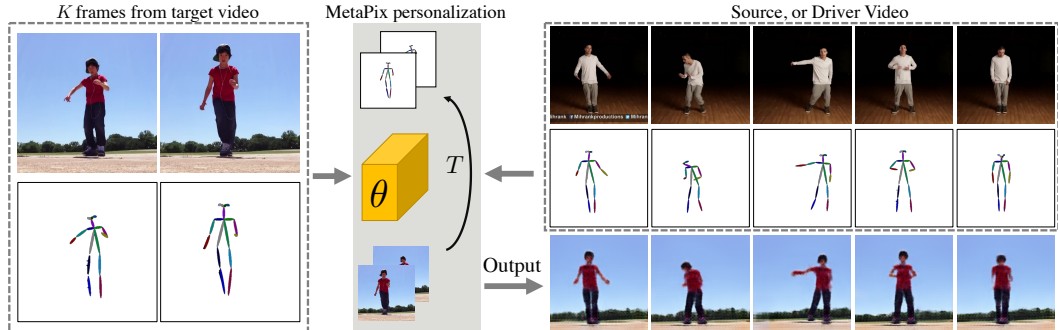

Figure 1: **Video retargeting on a budget.** Our goal is to retarget a source video to a target, *quickly* (running a few, $T$, iterations for adaptation) and *efficiently* (given a few, $K$, frames from the target domain). We achieve that by (meta)learning a model $\theta$ that is able to quickly and efficiently adapt to a given target video.

To that end, we propose MetaPix, a novel approach to personalization for video retargeting. Our formulation treats personalization as a few-shot learning problem, where the task is to *adapt* a generic generative model of human actions to a specific person given a few samples of their appearance. Our formulation is agnostic to the actual generative model used, and is compatible with both pose-conditioned transfer (Balakrishnan et al., 2018) or generative (Chan et al., 2018) approaches. Taking inspiration from the recent successes of meta-learning approaches for few-shot tasks (Nichol et al., 2018; Finn et al., 2017), we propose a novel formulation by adapting the popular first-order meta-learning algorithm Reptile (Nichol et al., 2018) for jointly learning initial weights for both the generator and discriminator. Hence, our model is optimized for efficient adaptation (personalization), given only a few samples and on a computational budget, and obtains stronger performance compared to a model not optimized in this form. Interestingly, we find this personalized model naturally enforces strong temporal coherence in the generated frames, even though it is not explicitly optimized for that task.

## 2 RELATED WORK

**Deep generative modeling.** There has been a growing interest in using deep networks for generative modeling of visual data, particularly images. Popular techniques include Variational Auto-Encoders (VAEs) (Kingma & Welling, 2014) and Generative Adversarial Networks (GANs) (Goodfellow et al., 2014). Particularly, GAN based techniques have shown strong performance for various tasks such as conditional image generation (Brock et al., 2019), image-to-image translation (Isola et al., 2017; Wang et al., 2018b; Zhu et al., 2017; Balakrishnan et al., 2018), unsupervised translation (Zhu et al., 2017) and domain adaptation (Hoffman et al., 2018). More recently, these techniques have been extended to video tasks, such as generation (Vondrick et al., 2016), future prediction (Finn et al., 2016) and translation (Bansal et al., 2018; Wang et al., 2018a). Our work explores generative modeling from a few samples, with our main focus being the task of video translation. There has been some prior work in this direction (Zakharov et al., 2019), though is largely limited to faces and portrait images.

**Motion transfer and video retargeting.** This refers to the task of driving a video of a person or a cartoon character given another video (Gleicher, 1998). While there exist some unsupervised techniques (Bansal et al., 2018) to do so, most successful approaches for articulated bodies involve using pose as an intermediate supervision. Recently, there have been two broad categories of approaches that have been employed for this task: 1) Learning to transform an image into another, given pose as input, either in 2D (Zhou et al., 2019; Balakrishnan et al., 2018; Siarohin et al., 2018; Ma et al., 2017) or 3D (Liu et al., 2018; Neverova et al., 2018; Walker et al., 2017). And 2) Learning a model to directly generate images given a pose as input (or, *Pose2Im*) (Chan et al., 2018). The former approaches tend to be more sophisticated, separately generating foreground and background pixels, and tend to perform slightly better than the latter. However, they typically learn a generic model across datasets that can transfer from a single frame, whereas the latter can learn a more holistic reconstruction by learning a specific model for a video. Our approach is complementary to such transfer approaches, and be applied on top of either, as we discuss in Section 3.

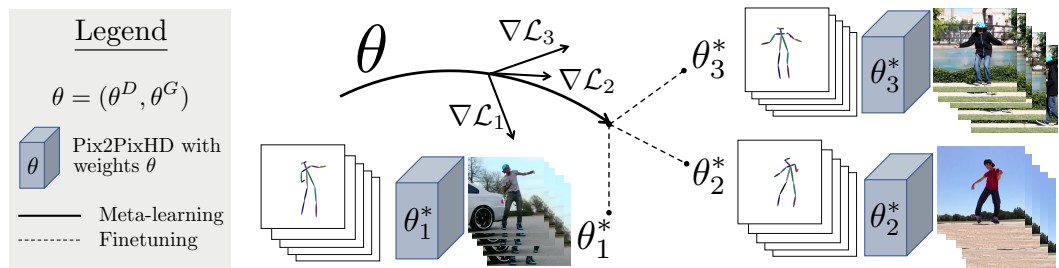

Figure 2: **Meta-learning for video retargeting.** Our goal is to learn a generic retargeting model $\theta$ (parameters of a Pix2PixHD (Wang et al., 2018b) in our case), such that it can be efficiently personalized for a specific person given only a few samples/frames of their appearance. We achieve it using meta-learning, where our model is optimized for personalization to a new person, given only $K$ samples of their appearance, and being trained for $T$ iterations.

**Few-shot learning.** Low shot learning paradigms attempt to learn a model using very small amount of training data (Thrun, 1996), typically for visual recognition tasks. Classical approaches build generative models that share priors across the various categories (Fei-Fei et al., 2006; Salakhutdinov et al., 2012). Another category of approaches attempt to learn feature representations invariant to intra-class variations by using hallucinated data (Hariharan & Girshick, 2017; Wang et al., 2018c) or specialized training procedures/loss functions (Wang & Hebert, 2016a; Bart & Ullman, 2005). More recently, it has been framed as a 'learning-to-learn' or a meta-learning problem. The key idea is to directly optimize the model, for the eventual few-shot adaptation task, where the model is finetuned using a few examples (Finn et al., 2017). Alternatively, it has also been explored in form of directly predicting classifier weights (Bertinetto et al., 2016; Wang & Hebert, 2016b; Wang et al., 2017; Misra et al., 2017).

**Meta Learning.** The goal of metalearning is to learn models that are good at learning, similar to how humans are able to quickly and efficiently learn to do a new task. Many different approaches have been explored to that end. One direction involves learning weights through recurrent networks like LSTM (Hochreiter et al., 2001; Santoro et al., 2016; Duan et al., 2016). More commonly, meta-learning has been used as a way to learn an initialization for a network, that is finetuned at *test time* on a new task. A popular approach in this direction is MAML (Finn et al., 2017), where the parameters are directly optimized for the test time performance of the task it needs to adapt to. This is performed by backpropagating through the finetuning process by computing second order gradients. They and others (Andrychowicz et al., 2016) have also proposed first-order methods like FOMAML that forego the need to compute second order gradients, making it more efficient at empirically small drop in performance. However, most of these works still tend to have the requirement of SGD to be used as the task optimizer. A recently proposed meta-learning algorithm, Reptile (Nichol et al., 2018), forgoes that constraint by proposing a much simpler first order meta learning algorithm that is compatible with any black box optimizer.

## 3 OUR APPROACH

We now describe MetaPix in detail. To reiterate, our goal is to learn a generic model of human motion, parameterized by $\theta$, that can quickly and efficiently be *personalized* for a specific person. We define speed and efficiency requirements in terms of two parameters: computation/iterations ($T$) and the number samples required for personalization ($K$), respectively. We now describe the base architecture, MetaPix training setup, and the implementation details.

**Base retargeting architecture.** We build upon popular video retargeting architectures. Notably, there are two common approaches in literature:1) Learning a transformation from one image to another, conditioned on the pose (Zhou et al., 2019; Balakrishnan et al., 2018) and 2) Learning a mapping from pose to RGB (Pose2Im), like (Chan et al., 2018). Both obtain strong performance and amenable to the speed and efficiency constraints we are interested in. For example in $K$-shot setting (i.e. to learn a model using $K$ frames), one can train the Pose2Im mapping using the $K$ frames in the former case, or use the $C_2^K$ pairs from $K$ frames to learn a transformation function from one of the

---

**Algorithm 1** Meta-learning for video re-targeting for the Pose2Im setup.

Initialize $\theta_D, \theta_G$ from pretrained weights
**for** iteration = $1, 2, ...$ **do**
    Sample $K$ pose image pairs from the same shot randomly
    Compute $\widetilde{\theta_D}, \widetilde{\theta_G} = \text{Pix2PixHD}_K^T(\theta_D, \theta_G)$, for $K$ images and $T$ iterations
    Update $\theta_D = \theta_D - \epsilon(\widetilde{\theta_D} - \theta_D)$
    Update $\theta_G = \theta_G - \epsilon(\widetilde{\theta_G} - \theta_G)$
**end for**

---

$K$ images to another in the latter case. They are also both compatible with our MetaPix optimization discussed next.

Pose2Im (Chan et al., 2018) approaches essentially build upon image-to-image translation methods (Isola et al., 2017; Wang et al., 2018b), where the input is a rendering of the body joints, and the output is an RGB image. The model consists of an encoder-decoder style generator $G$. It is trained using a combination of perceptual reconstruction losses (Johnson et al., 2016), implemented using an $L_1$ penalty over VGG (Simonyan & Zisserman, 2015) features and discriminator losses, where we train a separate discriminator network $D$ that is trained to differentiate the generated images from real images. The reconstruction loss forces it to be close to the ground truth, potentially leading to blurry outputs. Adding the discriminator helps fix that, as it forces the output onto the manifold of real images. Given its strong performance, we use Pix2PixHD (Wang et al., 2018b) as our base architecture for Pose2Im. For brevity, we skip a complete description of the model architecture, and refer the reader to (Wang et al., 2018b) for more details.

Pose Transfer (Balakrishnan et al., 2018; Zhou et al., 2019), on the other hand, takes a source image of a person and a target pose, and generates an image of the source person in that target pose. These approaches typically segment the limbs, transform their position as in the target pose, and generate the target image by combining the transformed limbs and segmented background by using a generative network like a U-Net (Ronneberger et al., 2015). These approaches can leverage learning to move pixels instead of having to generate color and background image from a learned representation. We utilize the Posewarp method (Balakrishnan et al., 2018) as our base Pose Transfer architecture due to available implementation.

**MetaPix.** MetaPix builds upon the base retargeting architecture by optimizing it for few-shot and fast adaptation for personalization. We achieve that by taking inspiration from the literature on few-shot learning, where meta-learning has shown promising results. We use a recently introduced first-order meta-learning technique, Reptile (Nichol et al., 2018). As compared to the more popular technique, MAML (Finn et al., 2017), it is more efficient as it does not compute a second gradient and is amenable to work with arbitrary optimizers as it does not need to backpropagate through the optimization process. Given that GAN architectures are hard to optimize, Reptile suits our purposes of its ability to use Adam (Kingma & Ba, 2015), the default optimizer for Pix2PixHD, as our task optimizer. Figure 2 illustrates the high level idea of our approach, which we describe in detail next.

We start with either a Pose2Im or a Pose Transfer trained base model. We then finetune this model as described in Algorithm 1. Note that Pix2PixHD is based on a GAN, so has two network weights to be optimized, the generator ($\theta_G$) and discriminator ($\theta_D$). In each meta-iteration, we sample a *task*: in our case a set of $K$ frames from a new video to personalize to. We then finetune the current model parameter to that video over $T$ iterations, and update the model parameters in the direction of the personalized parameters using a meta learning rate $\epsilon$. We optimize both $\theta_D$ and $\theta_G$ jointly at each step. Note that Posewarp employs a more complicated two-stage training procedure, and we metalearn only the first stage (which has no discriminator) for simplicity.

**Implementation Details.** We implement MetaPix for the Pose2Im base model by building upon a public Pix2PixHD implementation[1] in PyTorch, and perform all experiments on a 4 TITAN-X or GTX 1080Ti GPU node. We follow the hyperparameter setup as proposed in (Wang et al., 2018b). We represent the pose using a multi-channel heatmap image, and input and output are $512 \times 512$px RGB

---

[1]https://github.com/NVIDIA/pix2pixHD/

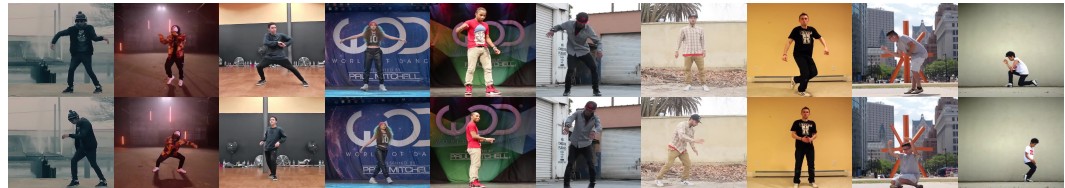

Figure 3: **Training data.** Frames from the additional training data we collected. We download 10 videos from YouTube, distinct from the ones used for personalization and evaluation.

images. The generator consists of 16 convolutional and deconvolutional layers, and is trained with a equally weighted combination of GAN, Feature Matching, and VGG losses. Initially, we pretrain the model on a large corpus of videos to learn a generic Pose2Im model as described in Section 4. During this pretraining stage, the model is trained on all of the training frames for 10 epochs using learning rate of 0.0002 and batch size of 8 distributed over the 4 GPUs. We experimented with multiple learning rates including $0.2, 0.02, 0.002$; however, we observed that higher learning rates caused the training to diverge. When finetuning for personalization, given $K$ frames and a computational budget $T$, we train the first $\frac{T}{2}$ iterations using a constant learning rate of 0.0002, and the remaining iterations using a linear decay to 0, following (Wang et al., 2018b). The batch size is fixed to 8, and for $K < 8$, we repeat the frames to get 8 images for the batch. For the metalearning, we set the meta learning rate, $\epsilon = 1$ with a linear decay to 0, and train 300 meta-iterations. We also experiment with meta learning rate, $\epsilon = 0.1$, however, was much slower to converge. To potentially stabilize metatraining, we experiment with differing numbers of updates to the generator and discriminator during iterations of Alg. 1, as well as simplified objective functions. Recall that the GAN loss adds significant complexity due to the presence of a discriminator that need also be adversarially finetuned. In total, our metalearning takes 1 day of training time on 4 GPUs. For the Pose Transfer base model, we apply MetaPix in a similar fashion on top of Posewarp[2], using the author provided pretrained weights. We will release the MetaPix source code for details.

## 4 EXPERIMENTS

We now experimentally evaluate MetaPix. We start by describing the datasets used and evaluation metrics. We then describe our base Pose2Im and Pose Transfer setup, followed by training that model using MetaPix. Finally, we analyze and ablate the various design choices in MetaPix.

### 4.1 DATASETS AND EVALUATION

We train and evaluate our approach on in-the-wild internet videos. Due to the lack of a standard benchmark for such retargeting tasks, we use the dataset as described in (Zhou et al., 2019) as our test set. These are a set of 8 videos downloaded from youtube, each 4-12 minutes long. We refer the reader to Figure 1 in (Zhou et al., 2019) for sample frames from this dataset. Additionally, we collect a set of 10 more dance videos from YouTube (distinct from the above 8), as our pre-training and meta-learning corpus. We provide the list of YouTube video IDs for both in the supplementary. Our models are only trained on these videos, and videos from (Zhou et al., 2019) are only used for personalization (using $K$ frames) and evaluation. Figure 3 shows sample frames from these newly collected videos.

**Evaluation and Metrics:** Similar to (Zhou et al., 2019), we split each of the 8 test videos into a training and test sequence in 0.85:0.15 ratio, and sample $K$ training and 2000 test frames from the test sequence. We use the same metrics as in (Zhou et al., 2019) for ease of comparison: Mean Squared Error (MSE), Structured Similarity Index (SSIM) and Peak Signal-to-Noise Ratio (PSNR). Each of these are averaged over the 2000 test frames from each of the 8 test videos. To compare our baselines and our method, pose retargeting as a task aims to minimize MSE and maximize SSIM and PSNR.

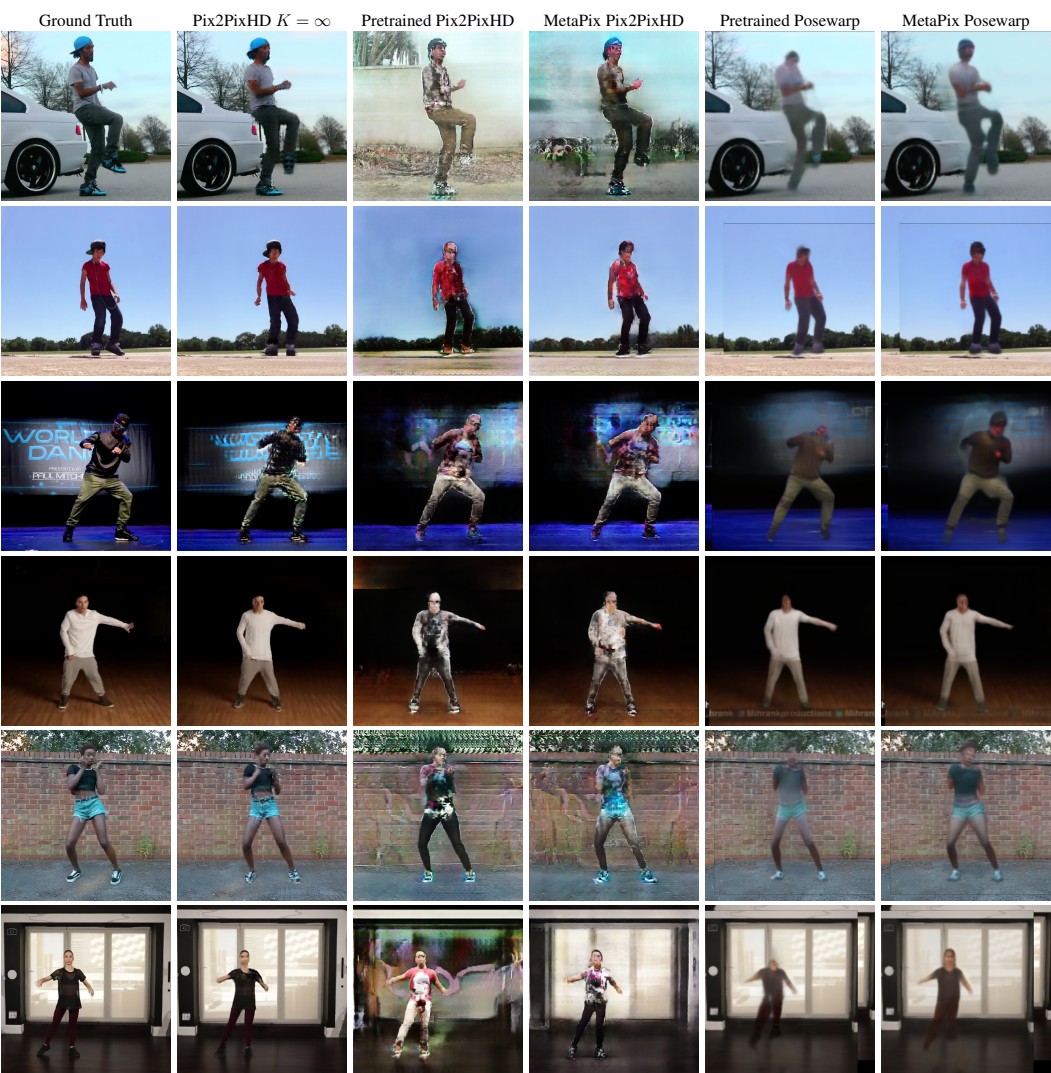

Figure 4: **Qualitative comparison.** Here we compare MetaPix with baselines and network architectures. The Pix2PixHD $K = \infty$ model is an upper bound tuned on *all* available frames for personalization. The last four columns compare the constrained setting, where at test time only $K = 5$ frames are used for personalization, over $T = 20$ iterations of fine-tuning. We visualize results of challenging pose-image test pairs that include rare poses not commonly scene in training. In this case, the pre-trained baseline tends to copy clothing and backgrounds from the training set, while MetaPix is much better at personalizing to the fine-tuning frames. The last two columns show that Posewarp is a more accurate base architecture than Pix2PixHD, but meta-learning on top of it still produces more accurate images with less blur.

Table 1: **Adding MetaPix.** We compare the Pix2PixHD and Posewarp models' performance. The models are initialized either at random, pretrained on the entire training set of videos, or meta-learned with MetaPix. Constraining the amount of frames and compute used at test time leads to a drop in performance as expected. In general, Pix2PixHD performs well with large training sets because it can learn to model subtle dependancies between pose and appearance. In contrast, Posewarp performs better with smaller training sets because it directly transfers pixels from the background and foreground. However, in both cases, meta-learning with MetaPix produces better performance, given the exact same test time constraints. We compare these methods qualitatively in Figure 4.

| Method | Init | $K$ | $T$ | SSIM | PSNR | MSE |
|---|---|---|---|---|---|---|
| Pix2PixHD | Random | $\infty$ | $\infty$ | 0.68 | 19.56 | $2,427.18$ |
| | Pretrain | $\infty$ | $\infty$ | 0.69 | 19.31 | $2,673.40$ |
| | Random | 5 | 20 | 0.08 | 9.51 | $7,801.19$ |
| | Pretrain | 5 | 20 | 0.35 | 12.00 | $5,576.28$ |
| | MetaPix | 5 | 20 | 0.39 | 13.73 | $4,696.00$ |
| Posewarp | Pretrain | $\infty$ | $\infty$ | 0.58 | 17.51 | $2,901.13$ |
| | Random | 5 | 20 | 0.40 | 12.25 | $4,670.81$ |
| | Pretrain | 5 | 20 | 0.55 | 16.53 | $3,140.91$ |
| | MetaPix | 5 | 20 | 0.56 | 16.94 | $2,962.67$ |

## 4.2 EVALUATING METAPIX

We start by building our baseline retargeting model, based on Pix2PixHD (Wang et al., 2018b; Chan et al., 2018). To get a sense of the upper bound performance of our model, we train the model for each test video with no constraints on $T$ or $K$, starting from the model pre-trained on our train set. Specifically, we use all the frames from the first 85% of each video, and train it for 10 epochs. We report the performance of this model in first section of Table 1 and show sample generations in second column of Figure 4. Since this model gets strong quantitative and qualitative performance, we stick with it as our base retargeting architecture through the rest of the experiments. We also employ a baseline retargeting model based on Posewarp for evaluation, but we focus on Pose2Im for further experimentation due to its relative simplicity.

Now we evaluate the performance of our model in constrained settings, where we want to learn to personalize given a few samples and in a constrained computational budget. Hence, we use a pretrained model on train set and a random model, and we personalize them by finetuning on each test video. As Table 1 shows, applying constraints leads to a drop in performance in all methods, as expected from using only 5 frames finetuned over 20 iterations. Finally, we compare that to the MetaPix model: in that case, we start from the pre-trained model, and do meta-learning on top of those parameters to optimize them for the transfer task as described in Section 3. That leads to a significant improvement over the pretrained model, showing the strength of MetaPix for this task.

In Figure 4[3], we visualize the predictions using the unconstrained model, as well as the constrained models trained using MetaPix and without, i.e. with simple pretraining. It is interesting to note that the meta-learned model is able to adapt to the color of the clothing and the background much better than a pretrained model, given the same frames for personalization. This reinforces MetaPix is a much better initialization for few-shot personalization than directly finetuning from a generic pretrained model. We further explore this quality of coherence in the next section.

## 4.3 ABLATIONS

We now ablate the key design choices in our MetaPix formulation. One of the strengths of our formulation is the explicit control on the supervision provided and computation the model is allowed to perform, and depending on the use-case, those parameters can easily be tweaked. We explore the effect of MetaPix on those parameters next on the Pose2Im base retargeting architecture.

**Variation in $K$:** We vary the amount of supervision for personalization, $K$, and evaluate its effect on the metrics in Figure 5. We compare the following models: a) Randomly initialized, b) Pretrained

---

[2]https://github.com/balakg/posewarp-cvpr2018
[3]Video visualization at https://youtu.be/NlUmsd9aU-4

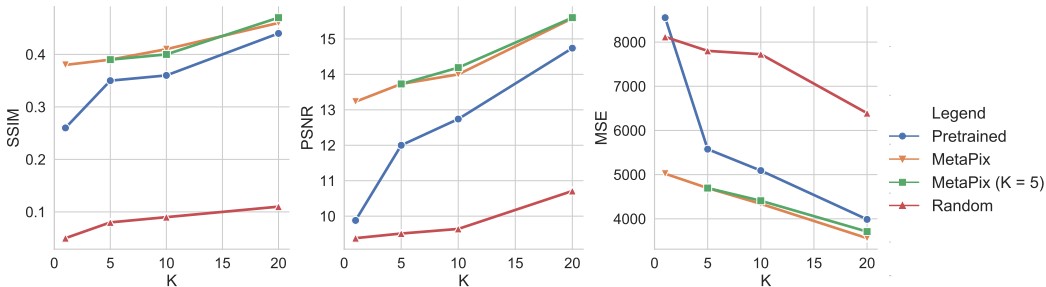

Figure 5: **Personalization using $K$ frames.** We find that while all initializations get better with increasing $K$, and using MetaPix consistently outperforms simple pretraining. Moreover we note that even using a model trained with MetaPix for $K = 5$ works well at any $K$ value used at test time, showing the generalizability of MetaPix. It is worth noting that the biggest gap is seen at lower values of $K$, showing our method is most useful in cases where one has little data for personalization.

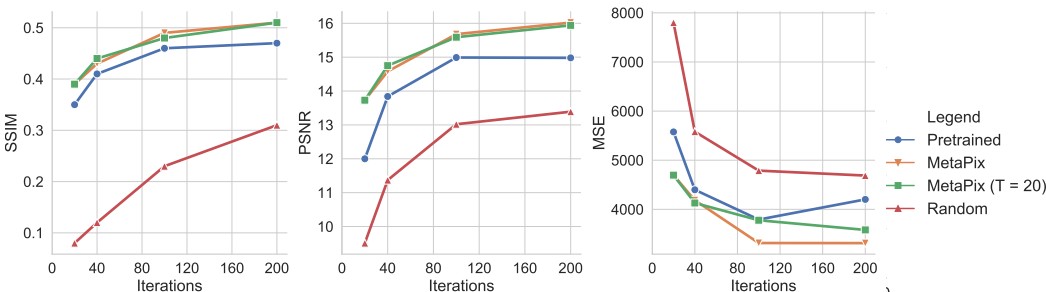

Figure 6: **Personalization after $T$ iterations.** We compare performance on increasing the computational budget for personalization. As expected all initializations improve with $T$, though MetaPix consistently outperforms random or pretraining. Again we see strong generalizability, as a MetaPix model trained for $T = 20$ performs well at other $T$ values used at test time.

on the train set, c) Trained using MetaPix for each value of $K$ and tested with the same $K$, and d) Trained using MetaPix for $K = 5$ and tested at each value of $K$. The last one tests the generalizability of MetaPix to different values of $K$ at train and test time. We find that the MetaPix trained models consistently perform better than a simple pretrained model on all metrics. Notably, the model only trained for $K = 5$ is still able to obtain strong performance at different $K$ values, showing the MetaPix trained model can generalize beyond the specific setup it is optimized for. The gap between the MetaPix trained model and the pretrained model tends to reduce with higher $K$, which is as expected: more data for personalization would likely reduce the importance of the initialization. However, there is a clear and significant gap for lower values of $K$, showing that MetaPix is highly effective for retargeting from few samples. In fact, we find that meta-learning is most effective for $K = 1$, corresponding to the challenging scenario of video-to-image retargeting.

**Variation in $T$:** Similar to variation in supervision, we experiment with varying the computation, or $T$, in Figure 6. We experiment with a similar set of baselines as in the case for $K$, and again observe that the MetaPix model consistently outperforms random initialization or pretraining on all metrics. Also, we see similar generalizability, as the model metatrained for $T = 20$ is able to perform well for other $T$ values at test time too. The ability for MetaPix to generalize across $K$ and $T$ implies cost-effective strategies for training. The computational cost for training a meta-learner is dominated by fine-tuning, which scales linearly with $K$ and $T$. Training with smaller values of both can result in significant speedups – up to $10\times$ in our experiments.

**Variation of meta learning rate $\epsilon$:** We also experimented with changing the meta learning rate. At $\epsilon = 0.1$ ($K = 5, T = 200$), we obtained SSIM=0.47, similar to what the pretrained model gets. Using our default $\epsilon = 1.0$, improves performance to 0.51. Hence, a higher meta learning rate was imperative to see improvements with MetaPix.

Pretrained                    MetaPix

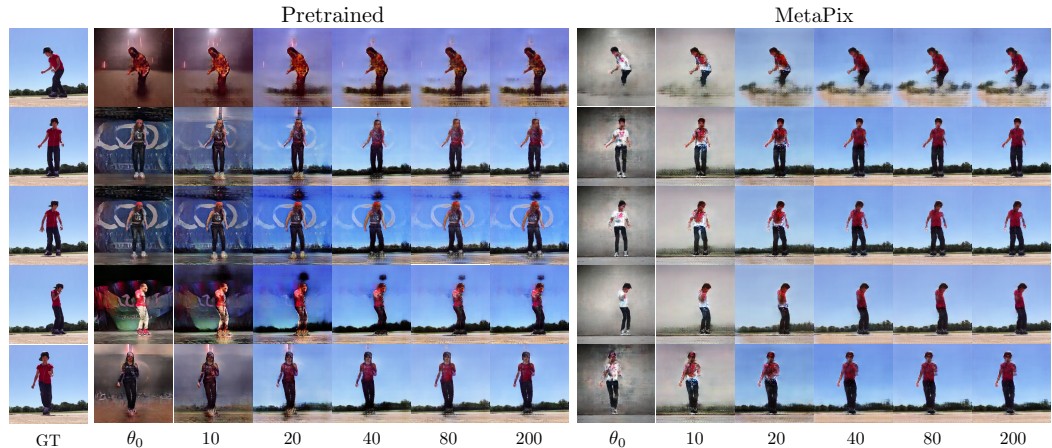

GT    $\theta_0$   10   20   40   80   200      $\theta_0$   10   20   40   80   200

Figure 7: **Visualizing finetuning between MetaPix and Pretrained.** We compare MetaPix's initialization for the $K = 5, T = 200$ task to our pretrained model initialization. We visualize models obtained during iterations of finetuning, at 0, 10, 20, 40, 80 and 200 iterations for 5 random test pose-image pairs. The images generated by MetaPix's initialization are temporally coherent, whereas the pretrained weights produce various training images depending on the pose. As observed in the intermediate iterations, the initialization translates its temporal coherence properties across finetuning iterations as well. This further reinforces our belief that MetaPix learns an initialization that is able to quickly adapt to the actor and background appearance from the few samples provided at test time.

**Only training the generator:** We apply Reptile in a GAN setting, where we jointly meta-optimize two networks. We also experimented with freezing one of the networks, specifically the discriminator, to the weights learned during pretraining. For our $K = 5, T = 200, \epsilon = 1.0$ setup, we obtain similar performance as optimizing both, suggesting that a 'universal' discriminator might suffice for meta-learning on GANs.

**Visualizing the dynamics of personalization:** In order to examine the process of personalization, we visualize models obtained during iterations of finetuning, at $10, 20, 40, 80$ and $200$ iterations for 5 random test pose-image pairs. We compare both the pretrained and metalearned model, trained for $k = 5, T = 200$. Figure 7[4] shows images generated by these intermediate iterations. Both methods learn clothing details and background colors after 20 iterations. Interestingly, MetaPix produces images that are temporally coherent, even upon initialization, while the pretrained baseline produces images whose background and clothing vary with pose. This more coherent initialization appears to translate to more coherent generated images after personalization.

## 5    CONCLUSION

We have explored the task of quickly and efficiently retargeting human actions from one video to another, given a limited number of samples from the target domain. We formalize this as a few-shot personalization problem, where we first learn a generic generative model on large amounts of data, and then specialize it to a small amount of target frames via finetuning. We further propose a novel meta-learning based approach, MetaPix, to learn this generic model in a way that is more amenable to personalization via fine-tuning. To do so, we repurpose a first-order meta-learning algorithm, Reptile, to adversarially meta-optimize both the generator and discriminator of a generative adversarial network. We experiment with it on in-the-wild YouTube videos, and find that MetaPix outperforms widely-used approaches for pretraining, while generating temporally coherent videos.

**Acknowledgements:** This research is based upon work supported in part by NSF Grant 1618903, the Intel Science and Technology Center for Visual Cloud Systems (ISTC-VCS), and Google.

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

# A   APPENDIX

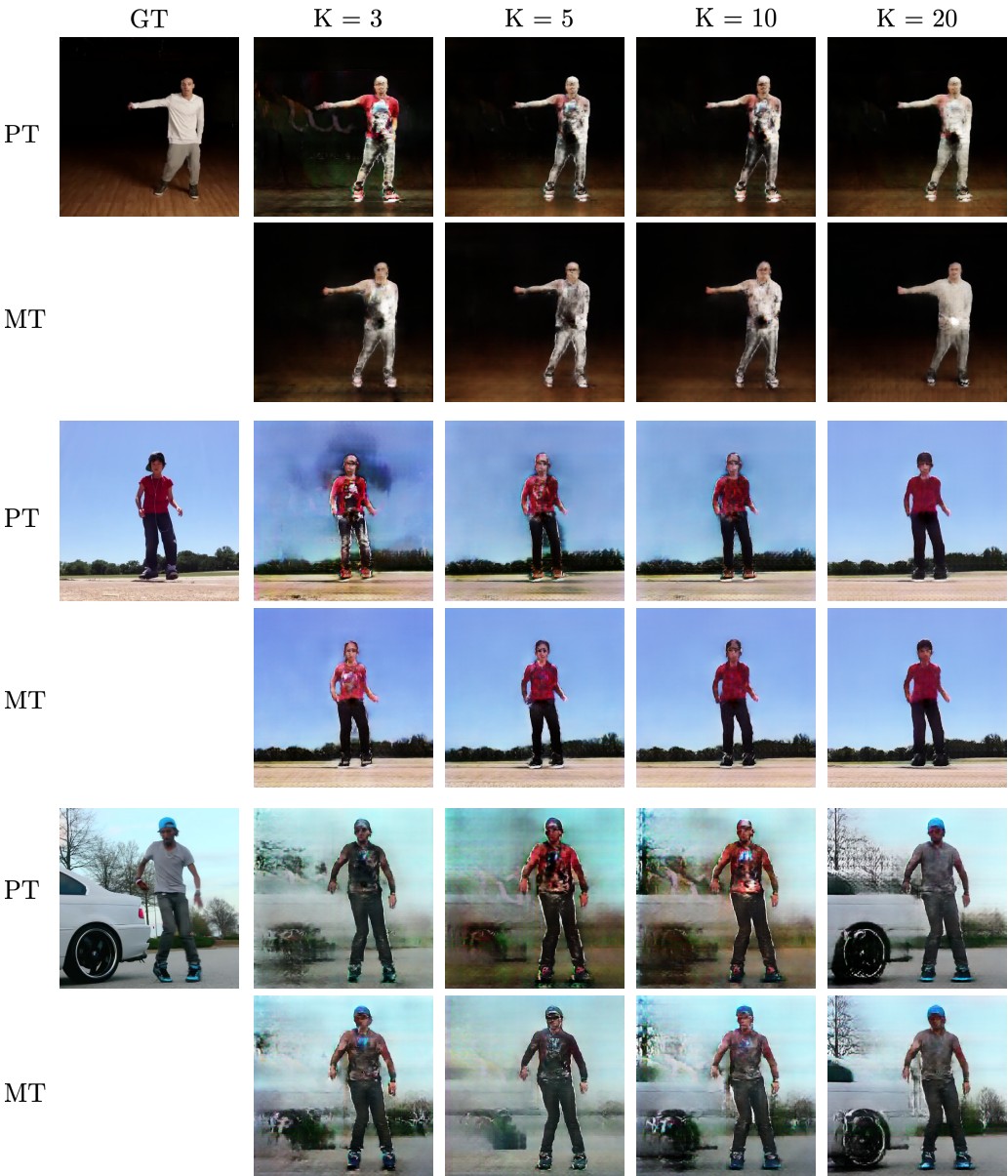

Figure 8: **Qualitative Variation in K.** We compare the MetaPix-trained models (MT) with their pretrained counterparts (PT) for $K = [3, 5, 10, 20]$. We fix the base architecture to Pix2PixHD and $T = 20$. With higher $K$, both methods generate good images, but with lower $K$, MetaPix generates backgrounds and clothing that better match the ground-truth. Our results illustrate that meta-learning excels in the few-shot regime.

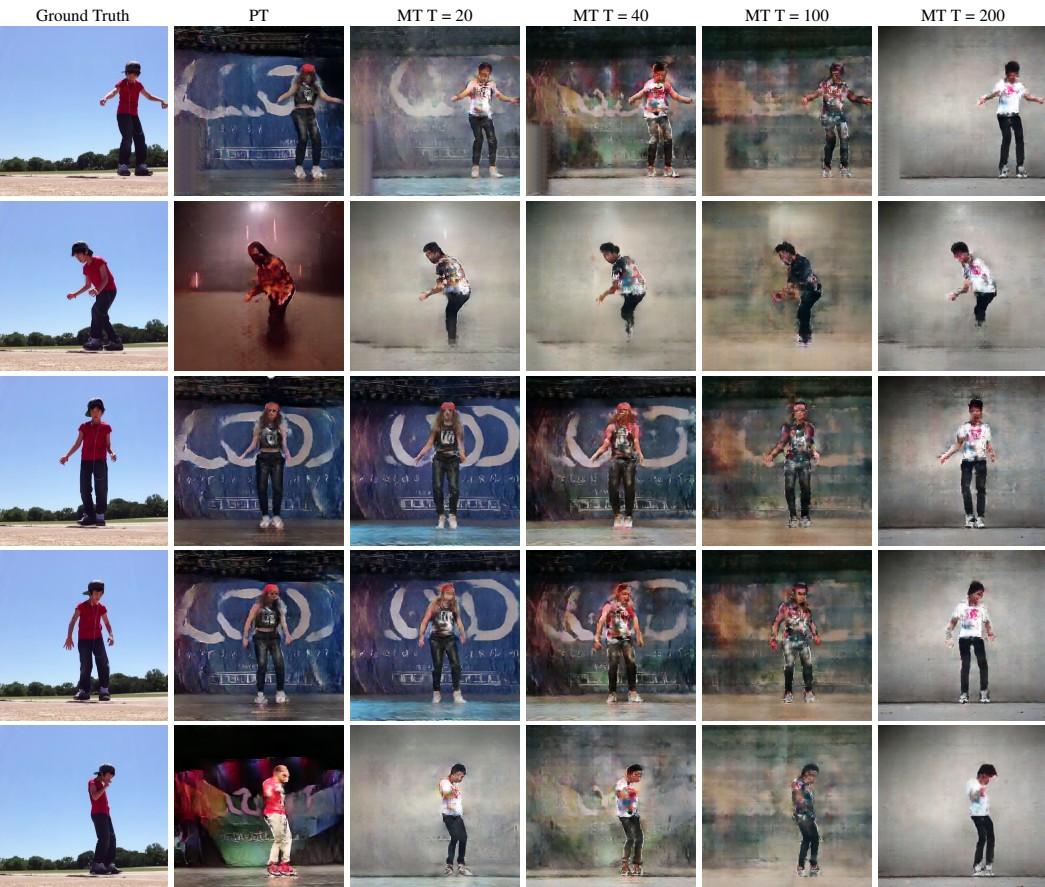

Figure 9: **Visualizing Initializations.**[a] We visualize knowledge captured by meta-learning by running the MetaPix-trained model (MT) without finetuning on a test video. We fix $K = 5$, vary $T = [20, 40, 100, 200]$. The pre-trained model (PT) generates an image from its training set. As we increase $T$, MetaPix learns to *factor* pose and appearance, generating consistent background and clothing appearance regardless of the human pose. Figure 7 demonstrates that such factored representations are easier to fine-tune, and result in more temporally stable generated videos.

---

[a]Video visualization is available at https://youtu.be/zFoT8VcbwsU

