# OpenReview forum: "MetaPix: Few-Shot Video Retargeting"
_ICLR.cc/2020/Conference — Accept (Poster)_

### Official Review · AnonReviewer3 · 2019-10-19
**Official Blind Review #3**

**Rating:** 6

**Review:**

In this paper, authors propose to address few shot video retargeting, where one should adapt a generic generative model of human actions to a specific person given a few samples of their appearance.

Overall, the paper is written with a good structure. I do like the problem setting and motivations in this paper. However, the solution is not quite novel for me. Both base model (Pix2PixHD) and few-shot adaptation (Reptile) come from the previous works. Their combination is somewhat incremental.

**Experience Assessment:**

I do not know much about this area.

**Review Assessment: Checking Correctness Of Derivations And Theory:**

I assessed the sensibility of the derivations and theory.

**Review Assessment: Checking Correctness Of Experiments:**

I assessed the sensibility of the experiments.

**Review Assessment: Thoroughness In Paper Reading:**

I made a quick assessment of this paper.

---

> ### Author Response · Authors · 2019-11-12
> **Response to Reviewer #3**
>
> We thank the reviewer for their feedback. We are glad that they find our problem setting and motivations in the paper interesting. With regards to our contribution, we think that our analysis on the application is insightful with regards to gaining an empirical understanding of Reptile. Leveraging the fact that we are learning an initialization of a generative model, we can “probe” the knowledge it has learned by generating images from its initialized state without any finetuning. Figure 9 shows meta-training learns to factor appearance from pose, ensuring that different pose inputs will generate consistent backgrounds and clothing cues. Then, we investigate the dynamics of specialization given K frames in Figure 7, which shows how the learned factored representation translates to more temporally coherent outputs. We refer the reviewer to our response to Reviewer 4 above where we further detail our contributions and link the video versions of Figure 7 and 9.

---

### Official Review · AnonReviewer1 · 2019-10-20
**Official Blind Review #1**

**Rating:** 6

**Review:**

This submission proposes an application of meta-learning to video frame generation modeling conditioned on human pose information, in order to allow the model to adapt to the context of each video. This context is provided in the form of a support set of K pairs of pose/frame images for the video. Reptile is used as the meta-learning method, and applied to two recently proposed video-frame generative networks (Pix2PixHD and Posewarp). In both cases, results show that Reptile is able to produce better adaptive models, i.e. models that when fine-tuned on the support set produce better image frames.

Though the originality of the work is somewhat weak (it's a relatively straightforward application of Reptile to Pix2PixHD and Posewarp), the problem setting is novel and I find the demonstration that Reptile works well in this setting interesting and valuable. The paper is also clearly written and easy to follow. For these reasons, I'm personally leaning towards recommending to accept this submission.

**Experience Assessment:**

I have published in this field for several years.

**Review Assessment: Checking Correctness Of Derivations And Theory:**

N/A

**Review Assessment: Checking Correctness Of Experiments:**

I carefully checked the experiments.

**Review Assessment: Thoroughness In Paper Reading:**

I read the paper thoroughly.

---

> ### Author Response · Authors · 2019-11-12
> **Response to Reviewer #1**
>
> We thank the reviewer for their feedback. With regards to the originality of our method, we believe that our analysis on Reptile is insightful and novel, especially the visualization of the initialization learned. Leveraging the fact that we are meta-optimizing a generative model, we are able to “probe” its knowledge simply by generating images from its initialized state without any fine-tuning. Figure 9 shows that meta-learning learns to factor appearance and pose, ensuring that different pose inputs will generate consistent backgrounds and clothing cues. Additionally, Figure 7 shows the dynamics of specialization, and shows meta-training leads to temporally coherent outputs. We also refer the reviewer to our response to Reviewer 4 above where we further detail our contributions and link the video versions of Figure 7 and 9.

---

### Official Review · AnonReviewer4 · 2019-11-01
**Official Blind Review #4**

**Rating:** 6

**Review:**

This paper proposes a novel and interesting task that learn to retarget human actions with few-shot samples. The overall pipeline is built by applying  meta-learning strategy on pre-trained retargeting module. It follows a conditional generator and discriminator structure that leverages few-shot frames to retarget the action of source video. The approach is technically sound.

The evaluations are compared with two baseline methods,  Pix2PixHD and Posewarp. Their evaluations are satisfactory and convincing, the results demonstrate some improvements over baseline models regarding both the selected metrics and the visualizing performance.

Though the proposed problem is novel and somewhat interesting, there are also several weaknesses of this work:
- The novelty of methodology is somewhat limited. It is more about merging several state-of-the-art modules in different tasks to tackle the few-shot retargeting problem. Though efforts may be needed to make the pipeline work, the overall contribution is not significant.
- The improvement obtained with proposed MetaPix module is not significant in few-shot setting according to Table 1. Additionally, could the authors provide some visualizing results for different K number, which would be interesting for analysis.







**Experience Assessment:**

I have read many papers in this area.

**Review Assessment: Checking Correctness Of Derivations And Theory:**

N/A

**Review Assessment: Checking Correctness Of Experiments:**

I carefully checked the experiments.

**Review Assessment: Thoroughness In Paper Reading:**

I read the paper at least twice and used my best judgement in assessing the paper.

---

> ### Author Response · Authors · 2019-11-12
> **Response to Reviewer #4**
>
> We thank the reviewer for their feedback. We are glad they find our problem setup novel and interesting, and experiments satisfactory and convincing. We address their concerns regarding novelty and significance of the approach here.
>
> - Novelty: While we agree that some of the tools we use (GANs, Reptile etc) have previously been studied, we believe the specific combination used for the novel problem setup we introduce is non-trivially different from any previous work. As Reviewer 1 also observed, we both empirically validate that existing meta-learning methods can work in this harder task setting, as well as investigate the qualities of the meta-learned initialization.
>
> Specifically, our paper provides more insight into what knowledge meta-learning can potentially capture. We are able to do so because we meta-learn a generative model, allowing us to “probe” its knowledge simply by generating images from its initialized state without any fine-tuning (akin to a zero-shot model). Figure 9, located in the Appendix, shows that meta-learning learns to factor appearance and pose, ensuring that different pose inputs will generate consistent backgrounds and clothing cues. On the other hand, simple pre-training memorizes specific combinations of pose and appearance present in the training set (first column in Figure 9). Specifically, our results illustrate that meta-learning discovers that factored representations of appearance and pose are easier to personalize for a target appearance. Moreover, we use the same image-generation technique to visualize the dynamics of what is being learned during iterations of fine-tuning. Our visualizations demonstrate that meta-learning for longer T ensures that fine-tuning is more stable, resulting in more temporally coherent outputs (Figure 7) with consistent backgrounds and clothing appearance regardless of the pose. Such temporal consistency is crucial for accurate video retargeting. Additionally, Figure 7 also compares the initial parameters learned by metalearning as opposed to pre-training, and we observe the metalearned features are more “generic” (eg, grey backgrounds), making them amenable to specialization. In summary, we argue that meta-learning of generative models opens up novel avenues for visualizations and (meta) model interpretation. We have also included video versions of Figure 7 (https://youtu.be/bxJJXCK4IoQ ) and Figure 9 (https://youtu.be/zFoT8VcbwsU ) to showcase the temporal stability of the metalearned initialization.
>
> Finally, our work also standardizes the problem of few-shot video generation with an in-the-wild realistic benchmark and strong baselines (code for all of which will be released). We believe this is a timely and interesting problem (as all reviewers agree), and our work would be critical in spurring further research in this area.
>
> - Significance: In the Pix2PixHD case, we see around a 4% improvement in SSIM, and 18% decrease in SSIM. Looking at the Figure 4 in the paper, we observe that the results are much closer to convergence, and that the background color is much closer to the ground truth than the initial feedback, meaning that we are closer to optimal solution beforehand. Therefore, our method both quantitatively and qualitatively improves upon the baselines.  We have additional video version of Figure 4 available to further display the qualitative improvement (https://youtu.be/NlUmsd9aU-4 ).
>
> - Additional visualizations: We have updated the Appendix of the paper to include some figures of meta-learned models in the setting K=3, K=5, K=10, and K = 20 for a fixed number of steps T=20 (Figure 8). We observe that the model converges in fewer epochs given more data, which is to be expected, since Reptile works best with a fewer samples in comparison to a pretrained initialization. Note that meta-learned initialization generates a qualitatively closer output to the ground truth, including the color of the background and the clothing detail on the person, faster than in the normal setting.

---

### Decision · Program_Chairs · 2019-12-19

**Decision:**

Accept (Poster)

**Comment:**

Three reviewers have assessed this paper and they have scored it 6/6/6 after rebuttal. Nonetheless, the reviewers have raised a number of criticisms and the authors are encouraged to resolve them for the camera-ready submission.